# Synergetic Design of Transparent Topcoats on ITO-Coated Plastic Substrate to Boost Surface Erosion Performance

**Xuan Zhang, Yuandong Chen, Wenqiao Zhang, Yanli Zhong, Pei Lei** 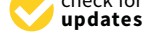**, Changshan Hao and Yue Yan ***

Beijing Engineering Research Center of Advanced Structural Transparencies to the Modern Traffic System, Beijing Institute of Aeronautical Materials, Beijing 100095, China; kaixuan1226@163.com (X.Z.); nishiyuandong@163.com (Y.C.); wqz_sunny@163.com (W.Z.); zhongyl3@163.com (Y.Z.); leipei.1232008@163.com (P.L.); dashan184504@163.com (C.H.)
* Correspondence: yue.yan@biam.ac.cn; Tel.: +86-010-6249-7662

**Abstract:** Transparent conductive films (TCFs) have received much research attention in the area of aeronautical canopies. However, bad wear, corrosion resistance and weak erosion performance of TCFs dramatically limit their scalable application in the next-generation aeronautical and optoelectronic devices. To address these drawbacks, three types of optically transparent coatings, including acrylic, silicone and polyurethane (PU) coatings were developed and comparatively investigated ex situ in terms of Taber abrasion, nanoindentation and sand erosion tests to improve the wear-resistance and sand erosion abilities of ITO-coated PMMA substrates. To elucidate the sand erosion failure of the coatings, the nanoindentation technique was employed for quantitative assessment of the shape recovery abilities under probe indentation. Results show that the PU topcoats can greatly enhance the sand erosion properties, which were superior to those of acrylic and silicone topcoats. This result can be attributed to the good toughness and self-healing properties of PU topcoats. Additionally, high hardness and good Taber abrasion properties of the ITO films and silicone topcoats did not have an obvious or affirmatory effect on the sand erosion abilities, based on their brittleness and irreparable properties under sand erosion.

**Keywords:** indium tin oxide (ITO) films; optically transparent coating; sand erosion; nanoindentation



## 1. Introduction

Transparent conductive films (TCFs), based on their relatively low electrical conductivity and high optical transmittance, are widely used in numerous fields, such as optoelectronic devices (smart windows, displays, etc.) [1–3], heatable layers in defrosting windows [4,5] and electromagnetic shielding and radar stealth in the aeronautical industry [6,7]. Among many TCF materials, the investigations of amorphous transparent conductive oxide films, such as conventional indium tin oxide (ITO) and amorphous indium zinc oxide (In$_2$O$_3$-ZnO, IZO) films [8,9], which are deposited on the polyester substrates, have received much attention, especially in the area of canopies in the aeronautical industry. However, conductive films, such as ITO and IZO, have numerous drawbacks such as bad wear, corrosion resistance and weak erosion performance [10–13], which can dramatically limit their large-scale application in the next-generation aeronautical and optoelectronic devices. To avoid these drawbacks, an optically transparent coating can be employed to be deposited on the conductive films to improve the durability of devices against marring, abrasion, erosion and a corrosive atmosphere, helpful for both the transparent conductive films and the transparent polyester substrates, such as poly(methyl methacrylate) (PMMA) [14], poly(ethylene terephthalate) (PET) [15] and polycarbonate (PC) [16].

Among many organic transparent coatings, the polyurethane (PU), acrylic and silicone coatings, have been used as transparent protective films for optical plastic lenses, computer monitors, keyboards and windshields of airplanes, based on excellent optical properties

and good construction performance [17,18]. PU coatings give the demanded flexibility, water resistance and improved scratch and corrosion resistance, and are widely used in the automotive and aeronautical industry [19]. Transparent acrylic coatings present a good compatibility with conductive films and PMMA substrates, but have worse anti-scratch properties [20]. Different to PU and acrylic coatings, silicone-based coatings with hard Si-O-Si net structures, provide higher hardness and impact resistance, which enhances crack prevention and scratch resistance [21]. The mechanical properties and scratch resistance of organic transparent coatings directly deposited on polyester substrates (PC, PMMA, PET, etc.) have been widely studied, based on traditional friction and wear experiments. However, reports on the comprehensive study of the structural, electrical, wear-resistance and sand erosion abilities of ITO films covered by organic transparent coatings are scarce. The assessment of the electrical and mechanical properties of organic coatings on ITO-coated polyester substrates is critical to the safe use and service life evaluation of ITO films.

This study aims to achieve transparent topcoats with good mechanical and strong anti-corrosive performance to protect the ITO films deposited on PMMA substrates. The ITO films were deposited by direct current magnetron sputtering at room temperature. Three kinds of optically transparent coatings, PU, acrylic and silicone coatings, were developed and deposited on the ITO-coated PMMA substrates. The mechanical and anti-corrosive properties of the topcoats on the ITO-coated PMMA substrates were systematically investigated and compared in terms of Taber abrasion, nanoindentation and sand erosion tests.

## 2. Experimental Procedures

### 2.1. Materials and Synthesis

Commercially available 3 mm thick PMMA sheets were rinsed with neutral detergent and distilled water, then dried at 70 °C for 60 min before surface pretreatment. The acrylate primer coatings, which can be effective to improve the adhesion of ITO films, were curtain-coated on PMMA sheets and cured at 85 °C for 2 h. The thickness of the acrylate primer coatings used in our experiment was approximately 3 μm. More details of preparation can be found elsewhere [21]. Deposition of the ITO thin films was performed by a direct current magnetron sputtering method at room temperature from a sintered conducting indium tin oxide ceramic target ($In_2O_3$:$SnO_2$ = 90:10 wt.%, $\Phi$70 mm × 5 mm). The distance of the substrate to the target was about 80 mm. The sputtering power was fixed at 100 w, which was low enough to produce no substantial substrate heating during sputtering. The substrate holder rotated with a velocity of 2 rpm to improve the uniformity. High purity Ar (30 sccm, standard cubic centimeter/min) and $O_2$ (1.2 sccm) were introduced through independent mass flow controllers after the vacuum chamber was evacuated below $3 \times 10^{-3}$ Pa and the working pressure was 0.30 Pa. ITO films with thickness of about 400 nm were obtained and the surface resistance was 20.0 $\Omega/\square$.

Three types of topcoats, including PU, acrylic and silicone coatings were curtain-coated on the ITO films. The details in the preparation of these topcoats has been presented elsewhere [21].

### 2.2. Characterization

Sand erosion performance of topcoats was determined from sand jet impingement erosion conditions. The sand impact velocity was between 20 and 125 m/s with sand sizes 50 μm. The jet impingement angle was set at 45 degrees, which is the similar flight erosion angle as a canopy. The investigative device is shown in Figure 1.

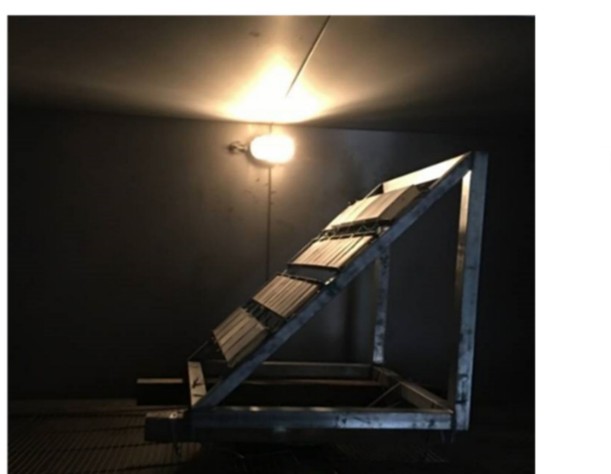
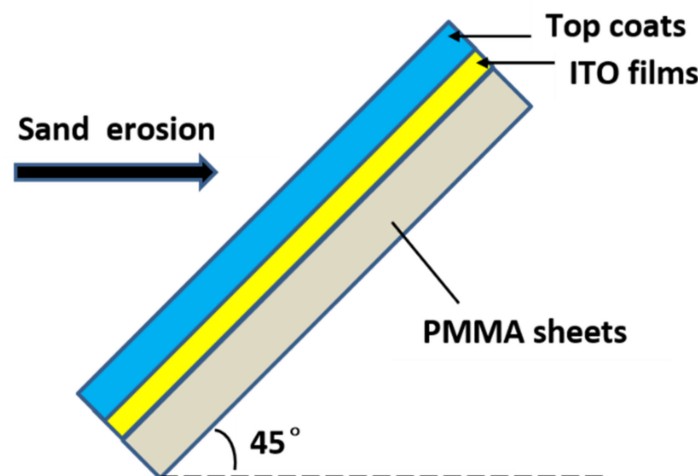

**Figure 1.** The picture of sand erosion device.

Nanomechanical properties were measured using TriboIndenter system (TI 950, Hysitron Inc., Minneapolis, MN, USA). The hardness was conducted by in indentation mode using a Berkovich, three-sided pyramid diamond indenter with an indenter tip of 1 μm radius of curvature. The cone angle was 143.2°. The ability to recover indentation damages was evaluated by means of morphological analysis by the TriboIndenter system.

The sheet resistance of the ITO films before and after sand erosion was measured by contactless resistivity systems (EC 80P, Napson Corp., Tokyo, Japan) based on eddy current principle. Ec 80p non-contact resistance measuring instrument is to emit electromagnetic waves to the measured material, and then detect the magnitude of the eddy current generated in the material to characterize the resistivity. The probe does not require tight contact materials for measurement and can be used for sheet resistance measurement of ITO films covered by organic coatings in this study. The abrasion resistance of transparent topcoats was determined by oscillating sand abrader (Taber 6160, Taber Industries, North Tonawanda, NY, USA) using the oscillating sand method (ASTM F735-17). The haze of transparent samples was detected by transmission haze meter (WGT-S, Shanghai Jingmi Industries, Shanghai, China).

## 3. Results and Discussion

### 3.1. Taber Abrasion Tests

Taber abrasion (ASTM F735-17) is a standard method to determine the resistance to surface abrasion of transparent plastics and coatings utilized in windows or viewing parts. Figure 2 shows the changes in haze of four kinds of transparent specimen (uncovered ITO films, ITO films covered by acrylic-based, silicone and PU topcoats) after different strokes using the oscillating sand. As shown in Figure 2, the haze increased with the increase in oscillation times, both for the uncovered ITO films and the three kinds of topcoats. Among these specimens, the acrylic-based topcoats showed the worst abrasion resistance and the haze markedly increased to almost 70% after 500 oscillations. Uncovered ITO films deposited on PMMA substrates had good abrasion resistance based on its high hardness, and the haze of uncovered ITO films increased from 0.5% to 7% after 500 oscillations, which was similar to that of the silicone and PU topcoats.

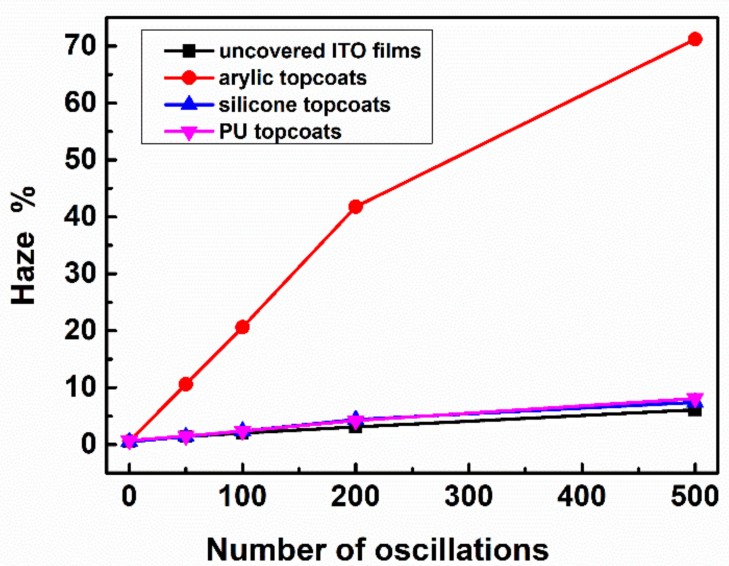

**Figure 2.** Haze changes of four kinds of transparent specimen (uncovered ITO films, ITO films covered by acrylic-based, silicone and PU topcoats) after different oscillation times.

*3.2. Nanoindentation Tests*

Nanoindentation tests using a depth-sensing method can provide nanomechanical properties of the thin coatings and films, which can avoid the substrate's influence [22]. Figure 3 shows the variation in hardness of the four kinds of transparent specimen (uncovered ITO films, ITO films covered by acrylic-based, silicone and PU topcoats) with contact depth under an increasing load between 10 μN and 1000 μN. As shown in Figure 3, uncovered ITO films represented the largest values of H based on the properties of inorganic conductive oxide films, which was consistent with the results of the Taber abrasion tests. The hardness level of the silicone topcoats was lower than that of the uncovered ITO films, whereas it was larger than that of the acrylic and PU topcoats. This was because of the characteristics of the Si-O-Si chemical group, which can be helpful for abrasion resistance properties. Among these topcoats, PU coatings with good flexibility and toughness, showed the lowest value of hardness. Different to the ITO films and silicone topcoats, the abrasion resistance of PU topcoats was not dependent on its hardness.

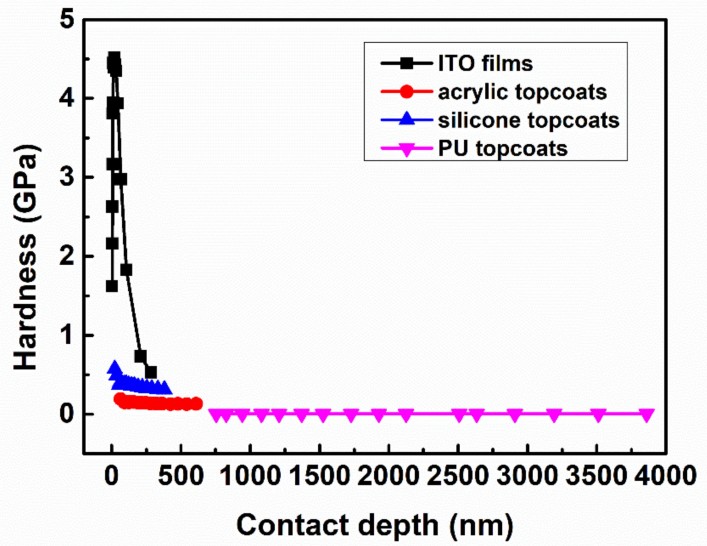

**Figure 3.** Hardness measurements of four kinds of specimen (uncovered ITO films, ITO films covered by acrylic-based, silicone and PU topcoats) at different indentation depths.

The wear resistance of thin films can also be demonstrated by the residual depth using indentation experiments. It was found that the shape recovery ability after indents was very important for films to resist the applied load. Figure 4a–c shows the surface images of uncovered ITO films and ITO films covered by acrylic-based and silicone topcoats under a certain load. Figure 4d represents the depth profiles after indentation. It was evident that the shape recovery effect did not exist for these three kinds of specimen, as there was no obvious difference for the residual depth right after and later after the indentation was found. As shown in Figure 4b, the acrylic-based topcoat was easy to be damaged and the residual depth was about 50 nm after indentation by a 200 μN load, whereas the uncovered ITO films presented the same residual depth after indentation by a 2000 μN load, as the ITO films were harder than the acrylic-based topcoat. For the silicane topcoat, the residual depth was less than 5 nm, which was much smaller than that of the ITO films after indentation by the same 2000 μN load.

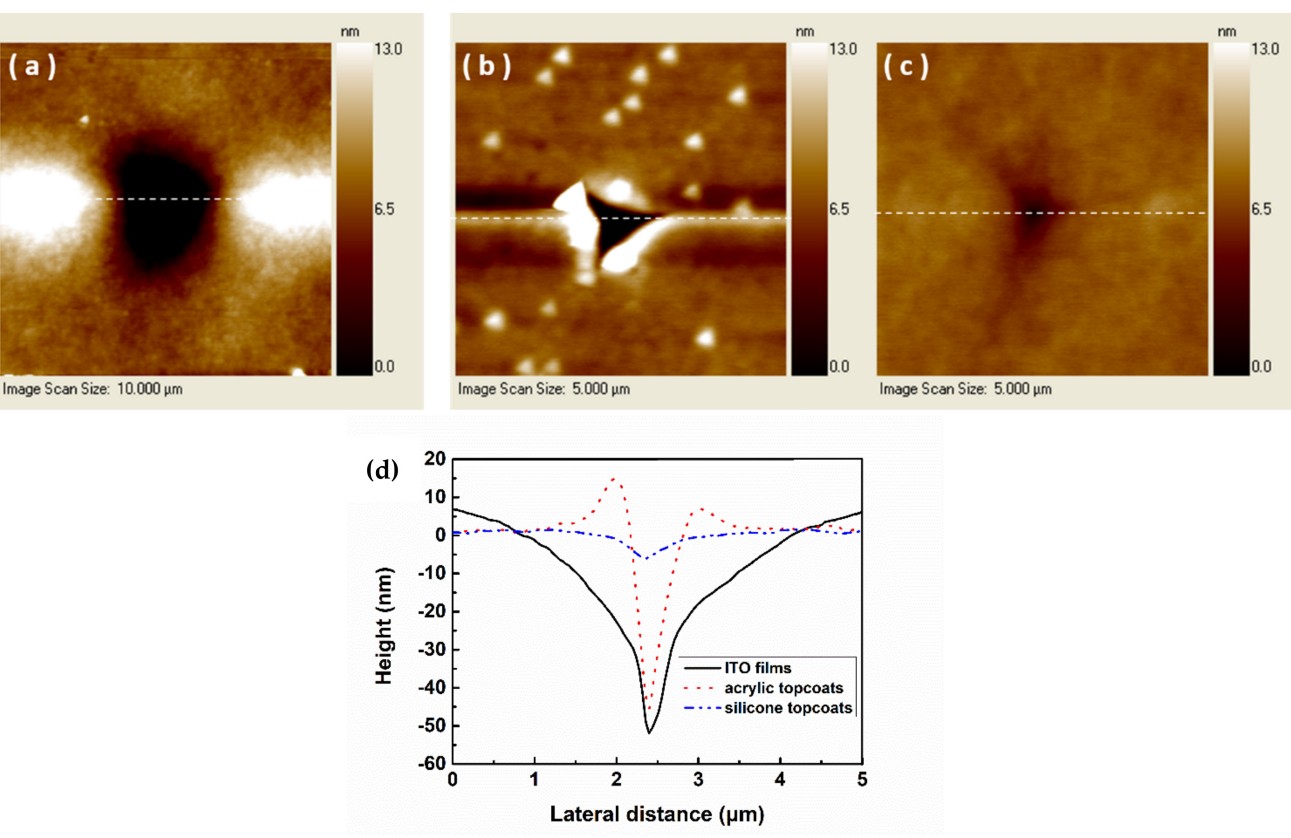

**Figure 4.** Surface images of (**a**) uncovered ITO films; ITO films covered by acrylic–based (**b**) and silicone (**c**) topcoats after indentation under 2000 μN, 200 μN and 2000 μN, respectively and (**d**) height profiles along the lines marked on the images.

The continuous scanning pattern of the indentation equipment was used to characterize the surface topographies of the PU topcoats after indentation, which can represent the ability to recover from damage. Figure 5a–d shows the surface images of the PU topcoat at different times after indentation by a 2000 μN load. As shown in Figure 5b,c, when the probe was scanning one part of the image, the other part to be scanned had partly recovered from the damage. Different to the acrylic-based and silicone topcoats, the PU topcoat demonstrated obvious self-healing properties, which can be characterized by the depth recovery ratio of indents in Figure 5e. As shown in Figure 5e, the residual depth decreased with the extension of recovery time. After 6 min, the residual depth changed from 61 nm to the undamaged stage, which confirmed the good ability to recover indentation damage. The recovery ability of the PU topcoats may play a dual role in affecting the sand erosion properties.

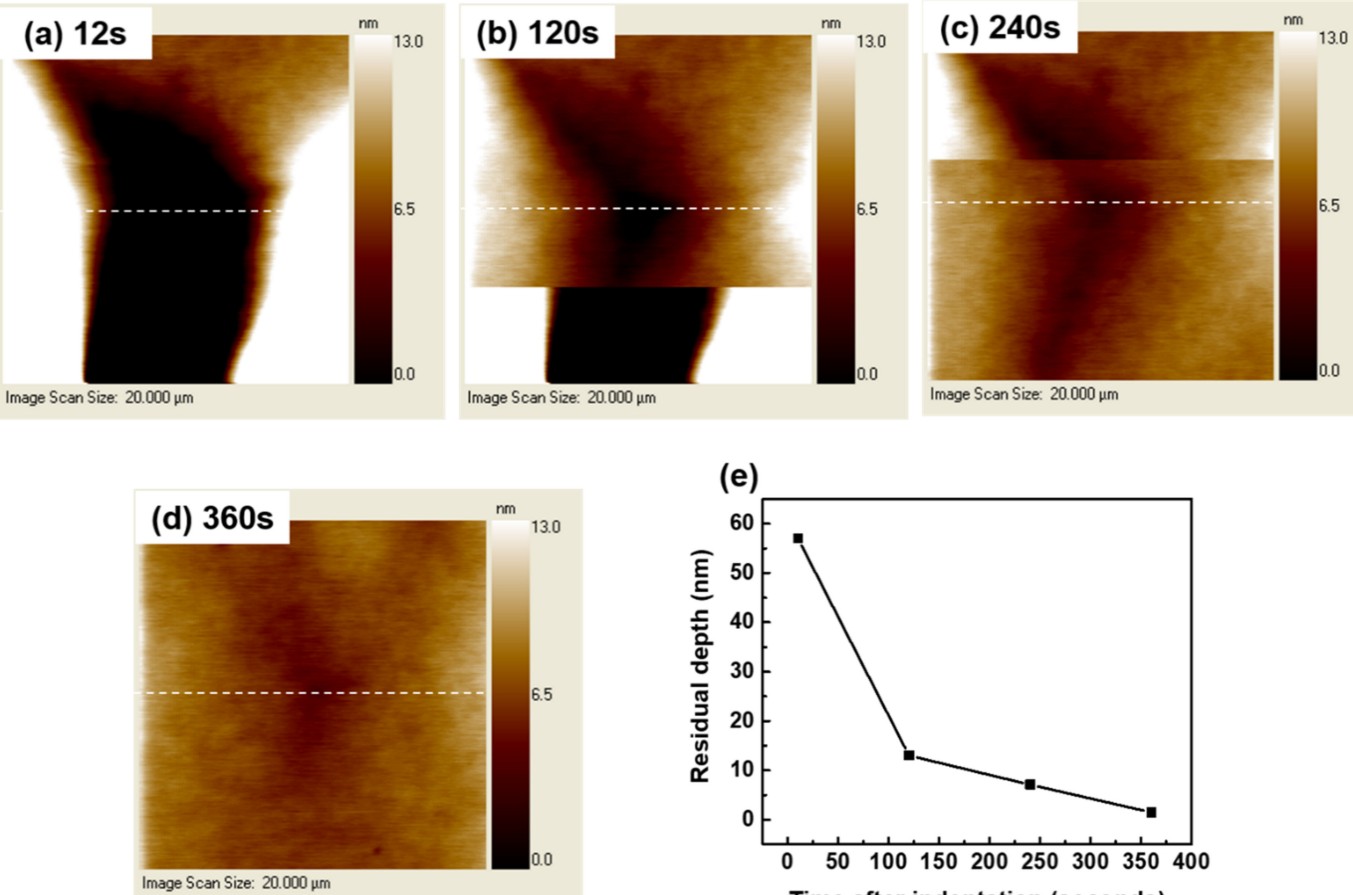

**Figure 5.** (**a**–**d**) Surface images of PU topcoat with different time after indentation and (**e**) height profiles along the lines marked on the images with different time.

### 3.3. Sand Erosion Tests

The microstructure, haze and surface resistance of these four kinds of transparent specimen (uncovered ITO films, ITO films covered by acrylic-based, silicone and PU topcoats) after sand erosion with sand impact velocities of 29 m/s using 50 μm sand as erodent, are characterized as shown in Figures 6–8, respectively. Both the ITO films and silicone topcoats presented a brittle fracture after sand erosion, as marked in the dotted area in Figure 6a,c. The ITO films below the silicone topcoats were easily exposed and damaged under the sand erosion because of the brittle fracture of the silicone topcoats. The acrylic-based topcoats were severely worn, as shown in Figure 6b, based on their poor wear resistance. However, ITO films below the acrylic-based topcoats were not exposed or damaged by sand erosion. There was no obvious abrasion damage for the PU topcoats and the ITO films below were well protected, as shown in Figure 6d.

Haze changes of these four kinds of specimen after sand erosion, as illustrated in Figure 7, were consistent with the optical morphology mentioned above. Similar with the results of the tabor abrasion tests of Figure 2, the acrylic-based topcoats presented as the largest worn and the haze was greatly increased by 77.1% after sand erosion for 90 min. Meanwhile, the haze of the uncovered ITO films and ITO films covered by silicone and PU topcoats was increased by 38.3%, 34.5% and 2.4%, respectively, after sand erosion for 90 min. The uncovered ITO films and silicone films showed obviously worse wear resistance after sand erosion, compared with the results of the PU topcoats, which demonstrated the opposite results of the Taber abrasion tests. Under the action of sand erosion, the damage mechanism was different.

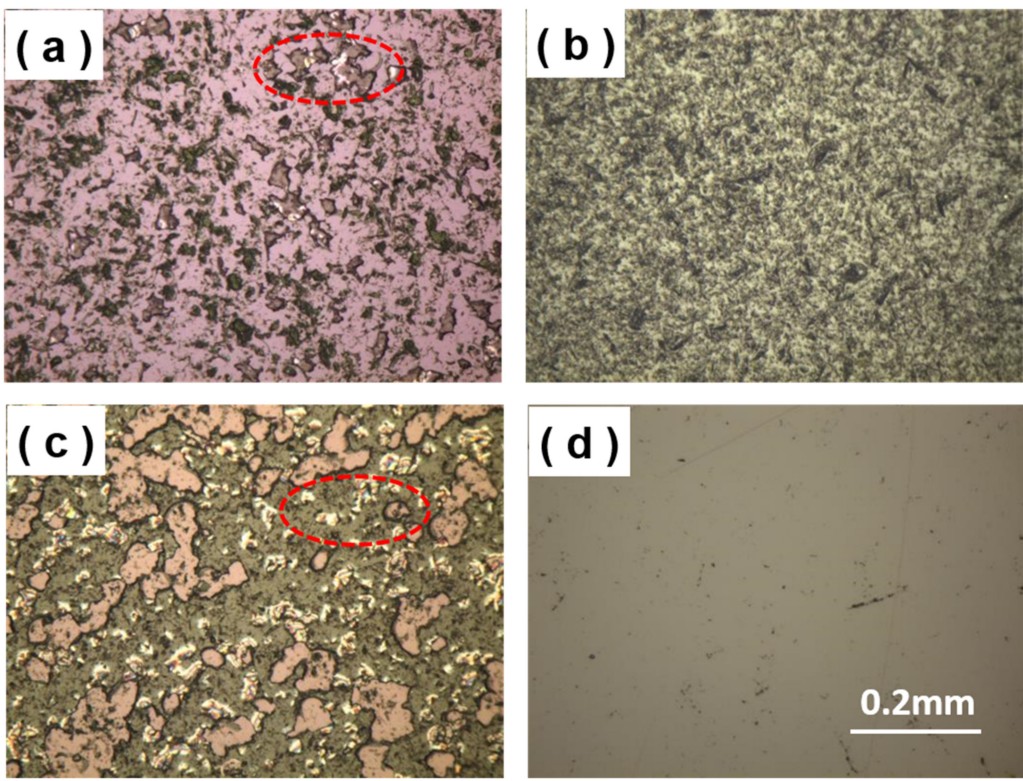

**Figure 6.** Optical microscope pictures showing (**a**) uncovered ITO films; ITO films covered by (**b**) acrylic-based, (**c**) silicone and (**d**) PU topcoats after sand erosion for 30 min.

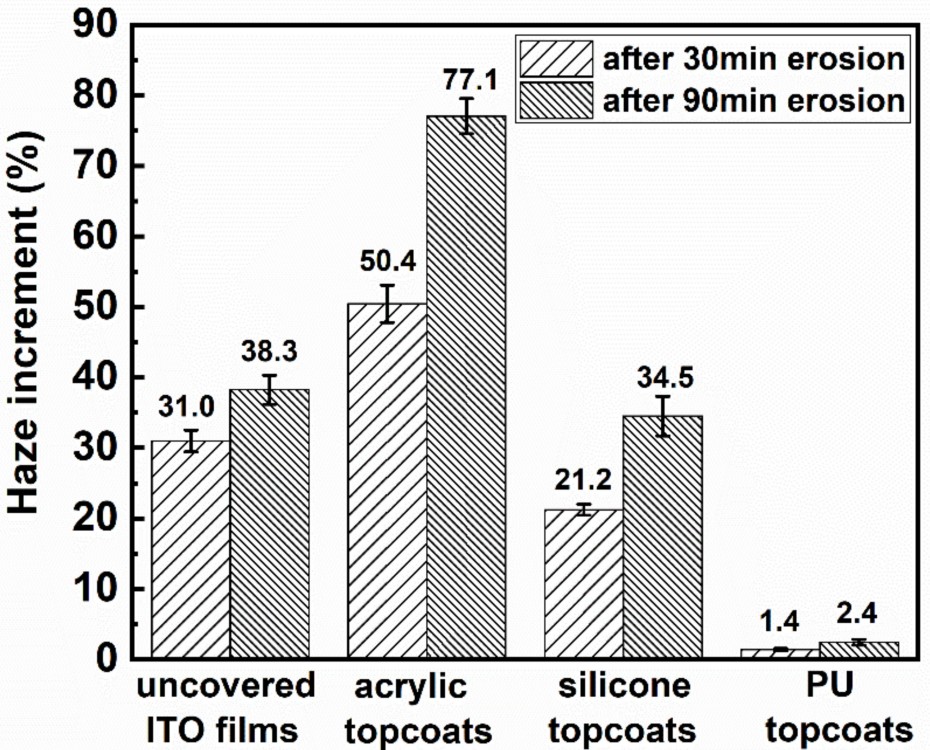

**Figure 7.** Variation in haze of uncovered ITO films, ITO films covered by acrylic-based, silicone and PU topcoats after sand erosion for 30 min and 90 min.

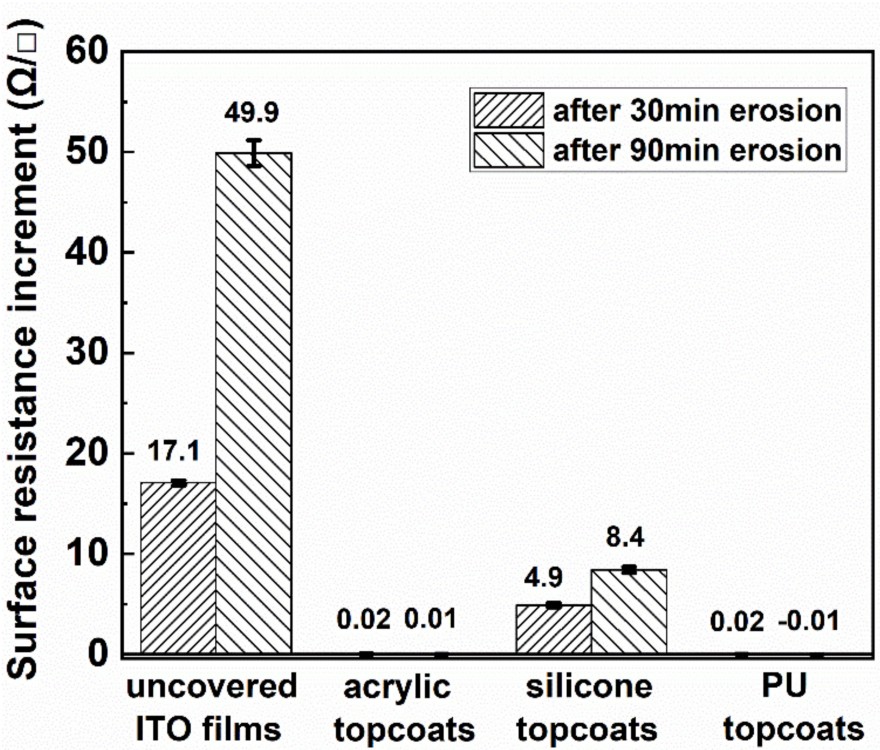

**Figure 8.** Variation in surface resistance of uncovered ITO films, ITO films covered by acrylic-based, silicone and PU topcoats after sand erosion for 30 min and 90 min.

The variation in surface resistance of ITO films without and with topcoats after sand erosion is presented in Figure 8. Evidently, the surface resistance of ITO films without topcoats and covered by silicone topcoats was increased with the time of sand erosion. The ITO films without topcoats were directly worn away by sand and the surface resistance was increased by 17.1 and 49.9 $\Omega/\square$ after sand erosion for 30 min and 90 min, respectively. Even under the protection of silicone topcoats, the surface resistance of ITO films was obviously increased, due to the brittle fracture of the silicone topcoats and the exposure of the lower ITO film under sand erosion, as proved by Figure 6c above. Otherwise, no difference was found for ITO films covered by acrylic and PU topcoats even after 90 min sand erosion, which indicated their good sand erosion abilities and properties.

The properties of topcoats, including mechanical and abrasion resistance, etc., are known to be the main reasons for their sand erosion abilities. Compared with the low hardness of PU topcoats from the nanoindentation tests, the high hardness of the ITO films and silicone topcoats does not have an obvious and affirmatory effect on the sand erosion abilities. Meanwhile, contrary to the results of the ITO films and silicone topcoats from the tabor abrasion tests, good abrasion resistance is also not the main affirmatory effect on the sand erosion abilities. Under the sand erosion, the brittleness and irreparable properties of the ITO films and silicone topcoats may play the main role in the failure of the coating, as proved by the nanoindentation images above. The PU topcoats, which presented very good self-healing properties, can overcome the compression indentation and enhance their sand erosion properties.

## 4. Conclusions

In summary, we developed three types of topcoats, acrylic, silicone and PU, which were deposited on ITO-coated PMMA substrates to improve the mechanical and sand erosion properties. The influence of topcoats on abrasion resistance, and mechanical and sand erosion properties was investigated according to Taber abrasion tests, and nanoindentation and sand erosion methods. Taber abrasion tests show that uncovered ITO films possessed similar abrasion resistance to the silicone and PU topcoats, based on the high hardness of

metal oxide films. Nanoindentation tests showed that the ITO films and silicone topcoats had higher hardness because of the ceramic ITO films and the Si-O-Si chemical groups, respectively. Contrary to this, the surface of the PU topcoats was very soft and presented the lowest values of hardness. Different to the irreparable properties of ITO films and silicone topcoats from the indentation tests, the PU topcoats presented very good self-healing properties, which can overcome the compression indentation and greatly enhance the sand erosion properties. Additionally, the good hardness and Taber abrasion properties of the ITO films and silicone topcoats did not have an obvious or affirmatory effect on the sand erosion abilities, based on their brittleness and irreparable properties under sand erosion. This synergetic design of transparent topcoats on ITO-coated PMMA substrate can be used to improve the durability of devices against marring, abrasion and an erosive atmosphere, which is useful and valuable for the scalable application of TCFs in the next-generation aeronautical and optoelectronic devices.

**Author Contributions:** Conceptualization and methodology, X.Z.; investigation, Y.C., W.Z. and Y.Z.; data curation, P.L. and C.H.; supervision, Y.Y. All authors have read and agreed to the published version of the manuscript.

**Funding:** This study is financially supported by the National Natural Science Foundation of China (No. 51702305).

**Institutional Review Board Statement:** Not applicable.

**Informed Consent Statement:** Not applicable.

**Data Availability Statement:** No new data were created or analyzed in this study. Data sharing is not applicable to this article.

**Conflicts of Interest:** The authors declare no conflict of interest.

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
