# Peer review of "Synergetic Design of Transparent Topcoats on ITO-Coated Plastic Substrate to Boost Surface Erosion Performance"

_coatings, doi:10.3390/coatings11121448_

Round 1
Reviewer 1 Report
This study develops three types of topcoats for ITO films, which are acrylic, silicone, and PU. This study conducts abrasion tests, nanoindentation, and sand erosion tests to characterize the mechanical properties of these topcoats. The work is innovative and solid. Here are some minor concerns: 1. It would be better to add statistical results and significance levels for Figure 7 and Figure 8, to convince your conclusion. 2. In the title of Figure 6, add an annotation for (d). 3. In the conclusion part, any potential application for these three topcoats?
Author Response
We sincerely thank referees for providing helpful comments for revision. A point-to-point reply to these comments is provided below.
This study develops three types of topcoats for ITO films, which are acrylic, silicone, and PU. This study conducts abrasion tests, nanoindentation, and sand erosion tests to characterize the mechanical properties of these topcoats. The work is innovative and solid. Here are some minor concerns:
Point 1: It would be better to add statistical results and significance levels for Figure 7 and Figure 8, to convince your conclusion.
Response 1: We have added the statistical results and significance levels suggested by reviewer into the revised manuscript, as shown in Fig. 7 and Fig. 8. Figure 7 related section: Similar with the results of tabor abrasion tests of Fig. 2, the acrylic-based topcoats presented the largest worn and the haze can be greatly increased by 77.1% after sand erosion for 90 min. Meanwhile, the haze of the uncovered ITO films and ITO films covered by silicone and PU topcoats were increased by 38.3%, 34.5% and 2.4%, respectively, after sand erosion for 90 min. Figure 8 related section: The ITO films without topcoats were directly worn away by sand and the surface resistance was increased by 17.1 and 49.9 Ω/â–¡ after sand erosion for 30 min and 90 min, respectively. Even under the protection of silicone topcoats, the surface resistance of ITO films was obviously increased, due to the brittle fracture of the silicone topcoats and the exposure of the lower ITO film under sand erosion, as proved by Fig. 6c above.
Figure 7. Variation in haze of uncovered ITO films, ITO films covered by acrylic-based, silicone and PU topcoats after sand erosion for 30 min and 90 min.
Figure 8. Variation in surface resistance of uncovered ITO films, ITO films covered by acrylic-based, silicone and PU topcoats after sand erosion for 30 min and 90 min.
Point 2: In the title of Figure 6, add an annotation for (d).
Response 2: As suggested by reviewer, the annotation for (d) has been added into the revised manuscript, as shown in Fig. 6.
Point 3: In the conclusion part, any potential application for these three topcoats?
Response 3: As suggested by reviewer, the potential application has been added in the revised manuscript, as shown in conclusion part. This synergetic design of transparent topcoats on ITO-coated PMMA substrate can be used to improve the durability of devices to against mar, abrasion and erosion atmosphere, which is useful and valuable for the scalable application of TCFs films in the next-generation aeronautical and optoelectronic devices.
Reviewer 2 Report
This paper reports on a comparative study of 3 different transparent coatings on an ITO-PMMA substrate to evaluate their effect on surface erosion performance and to improve the wear and erosion abilities of the substrate . These studies are particularly of high interest for applications in aeronautics and in optoelectronic devices.
20 references are listed all checked and mainly quoted in the introduction ;
Introduction well define the aim of this papers with appropriate references that allow understanding the choice of the 3 top coatings chosen for this study : acrylic, silicon and polyurethane ;
Experimental procedures section gives adequate details on the materials involved , sample preparation and characterisation : determination of sand erosion performance, nanomechanical properties and sheet resistance measurement.
Presentation and explanation and discussion of the results are well conducted with appropriate and clean figures.
The conclusion summarises the experimental study with the most significant results obtained allowing to consider the most effective topcoat;
Minor revision :
Figure 4 : missing d) on the figure
Figure 6 : in the legend: remind what is the dotted area; the magnification of this area would be appreciated to a better understanding of what is happening; and d) legend is missing ;
Optical microscopy views before erosion would also be valuable and show the impact of erosion on the surface;
In references :
Ref. 1 …. Chen , Z,F ; Marinova, V ; Petrova, D ….
Ref.2 … Abdel-Hady, K …
Ref. 3 … Souri, M. ; Park,S. ; Seo, A.
Sci. Rep.2020,10, 12486
Ref.7 : Nanoscale 2020, 12, 14589 – 14597
Ref . 16. Hwang, F. H.; FH, L. P. Cheng, L.P
Author Response
We sincerely thank referees for providing helpful comments for revision. A point-to-point reply to these comments is provided below.
This paper reports on a comparative study of 3 different transparent coatings on an ITO-PMMA substrate to evaluate their effect on surface erosion performance and to improve the wear and erosion abilities of the substrate. These studies are particularly of high interest for applications in aeronautics and in optoelectronic devices.
20 references are listed all checked and mainly quoted in the introduction ;
Introduction well define the aim of this papers with appropriate references that allow understanding the choice of the 3 top coatings chosen for this study: acrylic, silicon and polyurethane ;
Experimental procedures section gives adequate details on the materials involved, sample preparation and characterization: determination of sand erosion performance, nanomechanical properties and sheet resistance measurement.
Presentation and explanation and discussion of the results are well conducted with appropriate and clean figures.
The conclusion summarizes the experimental study with the most significant results obtained allowing to consider the most effective topcoat.
Point 1: Figure 4 : missing d) on the figure
Response 1: As suggested by the referee, “(d)” has been added into the revised manuscript (as shown in Figure 4).
Point 2: Figure 6 : in the legend: remind what is the dotted area; the magnification of this area would be appreciated to a better understanding of what is happening; and d) legend is missing ; Optical microscopy views before erosion would also be valuable and show the impact of erosion on the surface;
Response 2: The dotted area represents the typical morphology of brittle fracture of both the ITO films and silicone topcoats after sand erosion. As suggested by reviewer, both the reminder of the dotted area and the annotation for (d) has been added into the revised manuscript, as shown in part of Fig. 6.
Optical microscopy views before erosion were presented below. It can be seen that the surface of these four kinds of transparent specimen (uncovered ITO films, ITO films covered by acrylic-based, silicone and PU topcoats) was very smooth and no obvious damage defect can be found before sand erosion.
Figure. Optical microscope pictures showing (a) uncovered ITO films; ITO films covered by (b) acrylic-based, (c) silicone and (d) PU topcoats before sand erosion.
Point 3: In references : Ref. 1 …. Chen , Z,F ; Marinova, V ; Petrova, D ….
Ref.2 … Abdel-Hady, K …
Ref. 3 … Souri, M. ; Park,S. ; Seo, A.
Sci. Rep.2020,10, 12486
Ref.7 : Nanoscale 2020, 12, 14589 – 14597
Ref . 16. Hwang, F. H.; FH, L. P. Cheng, L.P
Response 3: As suggested by reviewer, all references above have been modified in the revised manuscript in part of references.
Reviewer 3 Report
Dear Authors,
I was pleased to read your manuscript about protection coating for conductive ITO. I have found the manuscript readily acceptable as is it. The topic that you bring up, and you discuss are worth of the global accession of Coating Journal. I believe that the results that you have shown are fruitfull for an extensive and continuative investigation on the field, and in the specific, for your idea on how to protect ITO. Moreover, the protection of ITO with transparent coating can be well utiized in the field of smart windows, and new kind of photovoltaic cells.
Thanks for the good results.
I have found only few details to clarify:
Line 41: avoid the “etc”
Line 43: avoid the “et al”. Can you add few references to the end of sentence? It seems you just drop knowledge without any clue.
Line 125: “H of silicone” what is H?
I wish you a good weekend.
Author Response
We sincerely thank referees for providing helpful comments for revision. A point-to-point reply to these comments is provided below.
Comments and Suggestions for Authors :
I was pleased to read your manuscript about protection coating for conductive ITO. I have found the manuscript readily acceptable as is it. The topic that you bring up, and you discuss are worth of the global accession of Coating Journal. I believe that the results that you have shown are fruitfull for an extensive and continuative investigation on the field, and in the specific, for your idea on how to protect ITO. Moreover, the protection of ITO with transparent coating can be well utiized in the field of smart windows, and new kind of photovoltaic cells.
Thanks for the good results.
I have found only few details to clarify:
Point 1: Line 41: avoid the “etc”
Response 1: As suggested by the referee, “etc” has been removed in the revised manuscript (as shown in Line 41).
Point 2: Line 43: avoid the “et al”. Can you add few references to the end of sentence? It seems you just drop knowledge without any clue.
Response 2: As suggested by the referee, “et al” has been removed and two references have been added in the revised manuscript (as shown in Line 43).
Point 3: Line 125: “H of silicone” what is H?
Response 3: H in the “H of silicone” means the hardness. In this paper, “H” is the short for “hardness”, which can be described above.
Reviewer 4 Report
Authors have coated different layers including acrylic, Silicone and PU on ITO coated PMMA substrate and studied the different properties. They confirmed that PU coating improve the surface erosion performance of ITO layer. This work is very interesting and presented in the manuscript. I have few comments, which are not mandatory to include in the manuscript but if authors wish they can include to improve the quality of the manuscript.
- It will be better if you include the curtain coating process schematic diagram in the manuscript. At least I would like to see.
- In the experimental section line 70-71: "The acrylate primer coatings were curtain-coated on PMMA sheets and cured at 85 °C for 2 h." What is this coating for? Is it same acrylic coating as top-coating? What is the thickness? Please describe it clearly and if it is mistake then remove from the manuscript.
- What was the ITO thickness?
- Can you please describe more or add reference for "contactless resistivity systems (EC 80P, Napson Corp, Japan )". I would like to know, if you add to manuscript that will be highly appreciated.
- Figure 5 (a, b, c) looks like defective image, can you please reconfirm the images?
Author Response
We sincerely thank referees for providing helpful comments for revision. A point-to-point reply to these comments is provided below.
Comments and Suggestions for Authors :
Authors have coated different layers including acrylic, Silicone and PU on ITO coated PMMA substrate and studied the different properties. They confirmed that PU coating improve the surface erosion performance of ITO layer. This work is very interesting and presented in the manuscript. I have few comments, which are not mandatory to include in the manuscript but if authors wish they can include to improve the quality of the manuscript.
I have found only few details to clarify:
Point 1: It will be better if you include the curtain coating process schematic diagram in the manuscript. At least I would like to see.
Response 1: The curtain coating process used in this study is one kind of flow coating and the liquid spreads under the action of gravity. The schematic diagram of coating process is shown in Fig.1 below. The size of sample used in this paper was about 140 mm×140 mm and we can hold the sample by hand. During the experiment, the sample was slightly inclined and a cup with a pointed tip can be used to pour the coating along the top of the sample. The coatings run down like a waterfall by gravity.
Figure 1. The schematic diagram of coating process.
Point 2: In the experimental section line 70-71: "The acrylate primer coatings were curtain-coated on PMMA sheets and cured at 85 °C for 2 h." What is this coating for? Is it same acrylic coating as top-coating? What is the thickness? Please describe it clearly and if it is mistake then remove from the manuscript.
Response 2: The acrylate primer coatings deposited on PMMA substrates is an effective method to improve the adhesion of ITO films, since the primer coatings can cover the scratches of PMMA and provide a clean and dense surface with lower roughness, which is beneficial to the deposition of ITO films. Effect of substrate pretreatment by primer layers on conductive films had been presented elsewhere [19]. The acrylate primer coatings used in this paper is same as the topcoats. The thickness of the acrylate primer coatings used in this study was approximately 3 μm. As suggested by reviewer, more details about the acrylate primer coatings have been added in the revised manuscript (as shown in part 2.1. Materials and synthesis).
Point 3: What was the ITO thickness?
Response 3: The thickness of ITO films was about 400 nm and the surface resistance was 20.0Ω/â–¡. As suggested by reviewer, more details about the thickness have been added in the revised manuscript (as shown in part 2.1. Materials and synthesis).
Point 4: Can you please describe more or add reference for "contactless resistivity systems (EC 80P, Napson Corp, Japan )". I would like to know, if you add to manuscript that will be highly appreciated.
Response 4: The sheet resistance of the ITO films before and after sand erosion was measured by contactless resistivity systems (EC-80P, Napson Corp, Japan) based on eddy current principle. With the normal 4-probe resistivity measuring instrument, the 4 probes must be tightly and well ohm connected with the conductive material during measurement, which is not applicable to detect conductive films with protective coatings in this study. EC-80p non-contact resistance measuring instrument is to emit electromagnetic waves to the measured material, and then detect the magnitude of the eddy current generated in the material to characterize the resistivity. The probe does not require tight contact materials for measurement and can be used for sheet resistance measurement of ITO films covered by organic coatings in this project. As suggested by reviewer, more details about the thickness have been added in the revised manuscript (as shown in part 2.2. Characterization).
Figure 2. Photos of the equipment.
Point 5: Figure 5 (a, b, c) looks like defective image, can you please reconfirm the images?
Response 5: As suggested by reviewer, we have reconfirmed the images of Fig. 5 and no problems have been found. Figure 5 (a, b, c) is the surface topographies by continuous scanning, which is used to characterize the coating's self-recovery ability after tip indentation. When the probe is scanning the upper part of the image, the lower part later to be scanned has been partly recovered, so the whole image may look like defective image.